# Quality of Sleep and Mental Symptoms Contribute to Health-Related Quality of Life after COVID-19 Pneumonia, a Follow-Up Study of More than 2 Years

**DOI:** 10.3390/biomedicines12071574

**Published:** 2024-07-16

**Authors:** Kathrine Jáuregui-Renaud, Davis Cooper-Bribiesca, José Adán Miguel-Puga, Yadira Alcantara-Calderón, María Fernanda Roaro-Figueroa, Mariana Herrera-Ocampo, Melodie Jedid Guzmán-Chacón

**Affiliations:** 1Unidad de Investigación Médica en Otoneurología, Instituto Mexicano del Seguro Social, Ciudad de México 06720, Mexico; cooper_2000@yahoo.com (D.C.-B.); adan.miguel@imss.gob.mx (J.A.M.-P.); 2Departamento de Psiquiatría, Hospital de Especialidades, Centro Médico Nacional Siglo XXI, Instituto Mexicano del Seguro Social, Ciudad de México 06720, Mexico; yadira.alcantara154@gmail.com (Y.A.-C.); jedid.chacon@gmail.com (M.J.G.-C.); 3Programa de Apoyo y Fomento a la Investigación Estudiantil, Facultad de Medicina, Universidad Nacional Autónoma de México, Ciudad de México 04510, Mexico; roaro.fer@gmail.com (M.F.R.-F.); maheoc.7@comunidad.unam.mx (M.H.-O.)

**Keywords:** COVID-19, sleep, health-related quality of life, mental health, cognitive performance

## Abstract

A follow-up study was designed to assess correlations among physical signs, quality of sleep, common mental symptoms, and health-related quality of life after moderate to severe COVID-19 pneumonia. Daily changes in dyspnoea and pulse oximetry were recorded (200 days), and four evaluations (in >2 years) were performed on quality of sleep, mental symptoms, cognitive performance, and health-related quality of life. In a single center, 72 adults participated in the study (52.5 ± 13.7 years old), with no psychiatry/neurology/chronic lung/infectious diseases, chronic use of corticosteroids/immunosuppressive therapy, or pregnancy. Daily agendas showed delayed decreases in dyspnoea scores compared to pulse oximetry and heart rate recordings; however, changes in pulse oximetry were minimal. Slight changes in cognitive performance were related to the general characteristics of the participants (obesity and tobacco use) and with the severity of acute disease (MANCOVA, *p* < 0.001). Health-related quality of life gradually improved (MANCOVA, *p* < 0.004). During recovery, bad quality of sleep and mental symptoms (mainly attention/concentration) contributed to the subscores on health perception and vitality in the health-related quality of life assessment. Early mental support services including sleep hygiene could be beneficial during rehabilitation after acute COVID-19.

## 1. Introduction

Globally, according to a conservative estimated incidence of 10% of infected people [1], at least 77 million people may have experienced post-COVID-19 conditions (>775,500,000 reported COVID-19 cases [2]). A systematic review and meta-analysis of 48 studies determined an estimated pooled global prevalence of post-COVID-19 conditions of 41.79% (95% C.I. 39.70–43.88%) [3], defined as “signs and symptoms that develop during or after an infection in line with COVID-19 that continue for >12 weeks and are not explained by an alternate diagnosis” [4]. In the United States of America, until March 2023, the online Household Pulse Survey of adults who ever experienced post-COVID-19 conditions showed a national prevalence estimate of 15.5% (95% C.I. 14.9% to 16.1%), affecting mainly middle-aged adults, including more females (19.0%, 95% C.I. 18.2% to 19.9%) than males (11.9%, 95% C.I. 11.1% to 12.7%) [5].

The clinical manifestations of post-COVID-19 conditions may vary among individuals [for review see [6]], and the most common symptoms reported to the WHO include fatigue, breathlessness, cognitive dysfunction, persistent cough, and chest pain [1]. A cross-sectional questionnaire-based survey (407 respondents/935 invited patients) showed 33.2%, 29.8%, and 5.7% of participants with at least one symptom at 1, 2, and 3 years after symptom onset or COVID-19 diagnosis, respectively, including fatigue, shortness of breath, cough, chest pain, palpitation, dysosmia, dysgeusia, hair loss, depressed mood, brain fog, loss of concentration, memory disturbance, and insomnia [7]. Data from 112 health care facilities of the United States of America (*n* = 17,487) showed that dyspnoea and chest pain were significantly elevated up to one-year post-infection, in comparison with other viral infections and control individuals [8]. A systematic review and meta-analysis showed that, at three months after COVID-19 diagnosis, 32% (95% C.I. 27% to 37%) of 25,268 patients (68 studies) reported fatigue and 22% (95% C.I. 17% to 28%) of 13,232 patients (43 studies) reported cognitive impairment [1].

Six months after acute disease, a retrospective study showed anxiety disorder in 17.4% of 236,379 patients, [9], while a cohort study showed post-traumatic stress disorder in 14%, 8%, and 9% of 251 patients at 1.5, 3, and 12 months after hospitalization due to COVID-19 [10]. Analysis on the incident sequelae of users of the Veterans Health Administration who survived for at least 30 days after COVID-19 diagnosis and were not hospitalized (*n* = 73,435) and users who did not have COVID-19 and were not hospitalized (*n* = 4,990,835) revealed an excess burden per 1000 patients after six months of 14.53 (95% C.I. 11.53–17.36) for sleep–wake disorders, 5.42 (95% C.I. 3.42–7.29) for anxiety and fear-related disorders, and 8.93 (95% C.I. 6.62–11.09) for trauma- and stress-related disorders [11].

Early in the pandemic, evidence showed that symptoms affecting mental functioning include impaired thinking, memory problems, troubles in finding words, and sleep disturbances, which may not correlate with hospitalization, treatment, viremia, or acute inflammation [12]. Factors contributing to mediate this variety of neuropsychiatric symptoms include persistent neuroinflammatory response, virus-induced hypercoagulable state, increased permeability of the blood–brain barrier with cytokine overload, direct viral neuronal invasion, systemic inflammation, as well as environmental and psychological factors [13,14]. A multicenter cohort study in the United Kingdom (*n* = 638) showed that anxiety and muscle weakness partially mediated the association between sleep disturbance and dyspnoea [15], while mood and sleep disturbances could be interpreted as part of the response to neuronal damage due to the pro-inflammatory state after infection [16]. A review of the literature on sleep disturbances related to COVID-19 supports that several factors may contribute to sleep dysfunction in patients hospitalized due to COVID-19, including duration of hospitalization, pre-existing concerns on mental health, psychological factors, and the individual immune response [for review [17]]. Moreover, evidence supports that a major complication of sleep loss is neuroinflammation, which may induce blood–brain barrier disruption [for review see [18]].

This study aims to fill gaps in the current literature on how a variety of idiosyncratic and clinical factors may influence the persistence of post-COVID symptoms. It was designed to assess the covariance of respiratory symptoms, quality of sleep, emotional and cognitive symptoms, and health-related quality of life through years after hospitalization due to COVID-19 pneumonia. The aims of this exploratory study were: 1. To assess daily changes in dyspnoea and pulse oximetry for 200 days, since at least six weeks after the onset of COVID-19 symptoms; and 2. To explore correlations among quality of sleep, mental health, cognitive performance, and health-related quality of life at four time points in a >2 year follow-up of adults who survived moderate to severe COVID-19 pneumonia, taking into account their general clinical characteristics.

## 2. Materials and Methods

After the institutional Research and Ethics Committees approved the research protocol to perform three evaluations, an amendment was authorized to perform a fourth evaluation (R-2020-785-157, amendment 31 January 2023). All of the evaluations were executed according to the Declaration of Helsinki and its amendments.

### 2.1. Participants

Seventy-two patients (mean age 52.5 years, range 25 to 85 years; 30 men/42 women) gave written informed consent to participate in the study, 6 to 50 weeks since the onset of COVID-19 symptoms. No sample size was calculated for the study. The patients were selected from a cohort of adults who had polymerase chain reaction (PCR)-confirmed SARS-CoV-2-infection and received standardized treatment for COVID-19 pneumonia as inpatients at a single center (Centro Medico Nacional sXXI, Instituto Mexicano del Seguro Social) between April 2020 and May 2022. According to the institutional records, 85 patients could fulfill the selection criteria. They were invited to participate by a personal phone call, describing the study protocol; however, six patients were not interested. Before the first psychiatric evaluation, informed consent was obtained from the remaining 79 candidates, with the opportunity to revoke their participation at any time. After psychiatric screening using the Mini-International Neuropsychiatric Interview [19] and the Neuropsychiatric Inventory [20], 74 patients fulfilled the selection criteria, but two died after the first evaluation. Then, 72 patients participated in the study. Three of the seventy-two participants did not attend the second evaluation (due to personal reasons) (*n* = 69) but they returned at the third (*n* = 72) and fourth (*n* = 72) evaluations, and they were included in the final sample (Figure 1). The preliminary evaluations confirmed that all 72 participants had no history of psychiatry/neurology/chronic lung/infectious diseases, or chronic use of corticosteroids/immunosuppressive therapy, or pregnancy. The general clinical characteristics of the 72 patients by gender are described in Table 1. Clinical information was retrieved from their institutional recordings, while information on tobacco and alcohol use was confirmed by direct interview with the patients and their relatives.

### 2.2. Evaluation Time Points

#### 2.2.1. Daily Agendas

The patients were instructed to record dyspnoea and pulse oximetry (with heart rate) on printed agendas, daily for 200 days.

#### 2.2.2. Questionnaires

Administration of questionnaires and face-to-face psychiatric interviews were performed four times since disease onset: at 143 ± 65 (mean ± standard deviation) days for the first evaluation, at 241 ± 66 days for the second evaluation, at 339 ± 75 days for the third evaluation, and at 857 ± 205 days for the fourth evaluation.

### 2.3. Procedures

#### 2.3.1. Daily Agendas

Patients were instructed to record their perceived dyspnoea according to the Borg Scale [21], and their pulse oximetry and heart rate on printed agendas. Individual reminders to complete the daily recordings and retrieval of pictures of the agendas were performed weekly, using the WhatsApp application for mobile phones. One patient was excluded from the analysis due to deficient completion of the recordings (52% of the recordings). Among the remaining 71 patients, 61 completed at least 90% of the recordings, and 10 completed 70% to 89% of the recordings. The recordings were performed at home, always using the same personal equipment (10 commercial brands), at circa the same time of day (coefficients of variation from 0.004 to 0.51 h). Since two patients provided sporadic daytime recordings, only 68 records were included for the estimate of daytime variation. According to the manufacturers of the oximeters, the pulse oximetry accuracy was ±3% to ±4%, and for heart rate it was 2%; however, one brand that was used by only one patient did not specify the accuracy.

**Figure 1 biomedicines-12-01574-f001:**
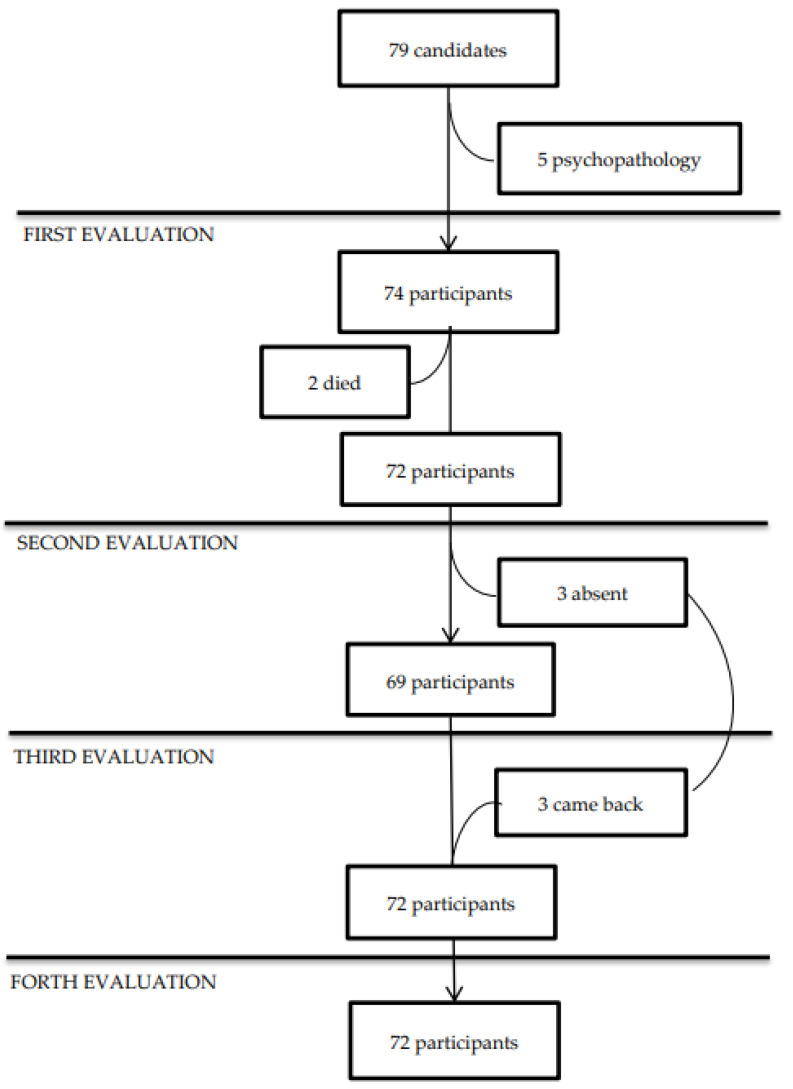
Flowchart of participants during the follow-up.

**Table 1 biomedicines-12-01574-t001:** Characteristics of the 72 patients participating in the study by gender, including general demographics and clinical characteristics. Mean and standard deviation (S.D.) with 95% confidence intervals (95% C.I.) are provided for numerical variables, and number and percentage (%) with 95% confidence intervals (95% C.I.) for categorical variables.

Variable	Men (*n* = 30)	Women (*n* = 42)	All (*n* = 72)
	Mean ± S.D. (95% C.I.)	Mean ± S.D. (95% C.I.))	Mean ± S.D. (95% C.I.)
Years of age	52.4 ± 12.2 (47.8–57.0)	52.5 ± 14.8 (47.9–57.2)	52.5 ± 13.7 (49.2–55.7)
Companion/no companion	20/10	32/10	52/20
Body mass index at hospital admission	32.1 ± 7.19 (29.4–34.0)	29.3 ± 5.1 (27.7–30.9)	30.5 ± 6.2 (29.1–31.8)
Body mass index at hospital discharge	31.5 ± 6.9 (29.0–34.1)	28.8 ± 5.0 (27.3–30.4)	29.9 ± 5.8 (28.6–32.2)
Days from symptom onset to hospitalization	8.1 ± 3.6 (6.7–9.4)	9.9 ± 4.3 (8.6–11.2)	9.1 ± 4.1 (8.2–10.1)
Days in hospital	11.6 ± 12.8 (6.7–16.4)	10.0 ± 7.4 (7.9–12.5)	10.8 ± 10.0 (8.4–13.1)
Oxygen saturation at hospital admission (%)	79.8 ± 12.6 (49.0–96.0)	83.1 ± 11.0 (79.6–86.5)	81.7 ± 11.7 (79.0–84.5)
Oxygen saturation at hospital discharge (%)	93.4 ± 2.6 (88.0–98.0)	93.9 ± 2.2 (93.2–94.6)	93.6 ± 2.4 (93.1–94.2)
Heart rate per minute at hospital admission	92.8 ± 19.7 (85.4–100.1)	98.0 ± 19.1 (92.0–104.0()	95.8 ± 19.4 (91.3–100.4)
Heart rate per minute at hospital discharge	80.8 ± 10.8 (76.7–84.8)	72.5 ± 10.6 (69.2–75.8)	75.9 ± 11.3 (73.2–78.6)
Leucocytes count at hospital admission	9.1 ± 4.8 (7.2–109)	10.1 ± 4.0 (8.8–11.3)	9.6 ± 4.3 (8.6–10.7)
Neutrophils at hospital admission	82.8 ± 10.5 (76.3–84.3)	85.3 ± 7.3 (83.0–87.6)	83.4 ± 9.0 (81.2–85.5)
Lymphocytes at hospital admission	13.7 ± 9.0 (10.3–17.2)	8.8 ± 4.7 (7.3–10.2)	10.7 ± 7.1 (9.0 12.4)
Leucocytes count at hospital discharge	8.0 ± 2.3 (7.1–8.9)	8.0 ± 2.4 (7.2–8.7)	8.0 ± 2.3 (7.4–8.5)
Neutrophils at hospital discharge	70.8 ± 14.5 (65.3–76.3)	72.7 ± 13.3 (68.5–76.9)	72.0 ± 13.6 (68.8–75.2)
Lymphocytes at hospital discharge	21.9 ± 11.8 (17.3–26.4)	18.6 ± 10.3 (15.4–21.9)	19.9 ± 10.9 (17.3–22.5)
	*n* (%, 95% C.I.)	*n* (%, 95% C.I.)	*n* (%, 95% C.I.)
Intensive care during hospitalization	6 (20%, 5.6–34.3%)	7 (16%, 4.9–27%)	13 (18%, 9.1–26.8%)
Systemic high blood pressure	14 (46%, 28.1–63.8%)	12 (28%, 14.4–41.5%)	26 (36%, 24.9–47.0%)
Type 2 diabetes	17 (56%, 38.2–73.7%)	13 (31%, 17.0–44.9%)	30 (41%, 29.6–52.3%)
Ever smokers	5 (16%, 2.8–29.1%)	11 (26%, 12.7–39.2%)	16 (22%, 12.4–31.5%)
Alcohol use	7 (23%, 7.9–38.0%)	24 (57%, 42.0–71.9%)	31 (43%, 31.5–54.4%)
Schooling			
<6 years	3 (10%, 0–20.7%)	1 (2%, -2.2–6.2%)	4 (5%, 0–10.0%)
6 years	4 (13%, 0–25.0%)	6 (14%, 3.5–24.4%)	10 (14%, 5.9–22.0%)
9 years	8 (27%, 11–42%)	13 (31%, 17.0–44.9%)	21 (29%, 18.5–39.4%)
12 years	8 (27%, 11–42%)	6 (14%, 3.5–24.4%)	14 (19%, 9.9–28.0%)
>12 years	7 (23%, 7.9–38.0%)	16 (38%, 23.3–52.6%)	23 (32%, 21.2–42.7%)

#### 2.3.2. Questionnaires

Before the face-to-face psychiatric interview by the same psychiatrist, the following questionnaires were administered:The Pittsburgh Sleep Quality Index [22], which comprises 19 items on 7 components: subjective sleep quality, sleep latency, sleep duration, habitual sleep efficiency, sleep disturbances, use of sleeping medication, and daytime dysfunction. A total score is calculated according to instructions by the developers [22]; a cut-off score > 5 is used to distinguish poor sleep, with Cronbach’s alpha coefficient ≥ 0.70 [23,24].The Hospital Anxiety and Depression Scale (HADS) [25], which comprises 14 items (7 for anxiety and 7 for depression) rated on a scale from 0 to 4. A total score is calculated by the sum of the ratings for all the items, and the two subscores are calculated by summing the ratings of the corresponding items. Cut-off scores of ≥8 for the subscores and ≥11 for the total score have shown sensitivity and specificity ≥ 0.70 [22,23], with a Cronbach’s alpha coefficient ≥ 0.67 [26], and test–retest reliability of the Spanish version > 0.85 [27].A Depersonalization/Derealization Inventory [28], which comprises 28 items rated on a scale from 0 to 4. A total score is calculated by summing all the individual scores (range 0 to 112), with a Cronbach’s alpha coefficient of 0.95 [28].The Dissociative Experiences Scale [29], which comprises 28 items on disturbances in memory, identity, and cognition, and feelings of depersonalization, derealization, absorption, and imaginative involvement. Scores on each item may range from 0% to 100% (using multiples of ten: 10%, 20%, 30%, etc.). A total score is calculated by dividing the sum of the individual scores by 28 (range 0% to 100%); a cut-off score of 8 is considered for the low normal range. The Spanish version has a Cronbach alpha coefficient of 0.96 [30].The Post-traumatic Stress Disorder Symptom Severity Scale-Revised by Echeburúa (2016) [31], which is a 21-item structured interview based on DSM-5 criteria, with a Cronbach alpha coefficient of 0.91 [31]. Individual diagnosis was performed according to both the scale and face-to-face psychiatric interview.The Montreal Cognitive Assessment (MoCA) [32], which comprises six domains: Visuo-spatial, Naming, Attention, Language, Abstraction, Memory, and Orientation. The items in each domain yield individual index scores, with a maximum total score of 30 points, where a 24/25 cut-off has shown higher specificity than a 25/26 cut-off [33]. It has shown a Cronbach’s alpha coefficient > 0.70 [34], with the reliability of the Spanish version of 0.89 [35].The Short-Form Health Survey (SF-36) [36], which is composed of 36 items on eight domains of health-related quality of life: physical function, social functioning, role limitations due to physical problems, role limitations due to emotional problems, mental health, vitality, pain, and general health perception. Scores on each domain range from 0 to 100 [37]. The Spanish version has a Cronbach’s alpha coefficient > 0.7 for all dimensions, except for social functioning (alpha coefficient = 0.45) [38].

### 2.4. Statistical Analysis

Statistical analysis was performed according to the data distribution, after the Kolmogorov–Smirnov test. Bivariate analysis included a “*t*” test for independent samples to compare data by gender and according to the cut-off points of the scores on quality of sleep, anxiety/depression, dissociative experiences, and cognitive performance. Repeated measures analysis of variance with Duncan’s test or the Friedman test was used to compare the repeated measures performed at the four evaluations, and the Spearman coefficient of correlation was used to assess simple correlations on the scores of the questionnaires at each evaluation. According to the bivariate analysis, using a general linear model, two repeated measures multivariate analyses of covariance were performed selecting the repeated measures on cognitive performance as the outcome variable for the first analysis and the repeated measures on health-related quality of life for the second analysis. The analyses were performed with a statistical significance of 0.05.

## 3. Results

### 3.1. Descriptive Statistics and Exploratory Bivariate Analyses

#### 3.1.1. Agendas

Figure 2 shows the 10-day median/mean set summaries of the dyspnoea score/pulse oximetry and heart rate recordings, respectively. A decreasing trend in dyspnea supports the attenuation of symptom severity over time.

Comparison among the first 10 days and every 50 days sets showed that, compared to men, women had higher oxygen saturation at the first 10 days, 11–50 days, and 151–200 days (“*t*” test, t-values from 1.994 to 2.448, *p* ≤ 0.05) and lower heart rates during 51 to 100, 101–150, and 151–200 days (“*t*” test, t-values from 1.992 to 2.393, *p* ≤ 0.05). Obesity was related to lower oxygen saturation during the first 10 days and 11–50 days (“*t*” test, t = 2.314 and 2.809, *p* ≤ 0.03), while diabetes mellitus was related to lower oxygen saturation during almost all of the follow-up, except for the first 10 days (“*t*” test, t-values from 2.061 to 2.414, *p* ≤ 0.05), and to higher heart rate during 101–150 and 151–200 days (“*t*” test, t-values from 2.364 to 2.669, *p* ≤ 0.03). No differences were observed according to tobacco use or the need of intubation for mechanical ventilation during hospitalization.

Repeated measures analysis, including the time since disease onset as a covariate, showed, compared to the first 10 days, a significant decrease in dyspnoea scores at 101–150 days (MANOVA and Duncan’s test, F = 7.586, *p* < 0.00001; and Friedman test X^2^ = 71.366, *p* < 0.00001), with no further change afterwards, while a minimal increase in pulse oximetry (MANOVA and Duncan’s test, F = 20.278, *p* < 0.00001) with a decrease in heart rate (MANOVA and Duncan’s test, F = 9.902, *p* < 0.00001) were evident at 51–100 days, with no significant change afterwards.

#### 3.1.2. Questionnaires

Questionnaires with face-to face psychiatric interviews. Scores at each of the four evaluations are described in Table 2. Compared to men, women had higher scores on sleep quality (worse sleep) at the four evaluations (“*t*” test, t-values from 2.577 to 3.540, *p* < 0.02) and lower scores on health-related quality of life at the first and second evaluations (“*t*” test, t-values = 2.792 and 2.954, *p* < 0.02). Compared to non-smokers, at the four evaluations, smokers had higher scores on sleep quality (worse sleep) (“*t*” test, t-values from 2.577 to 2.863, *p* < 0.02), on the HADS total score (“*t*” test, t-values from 2.439 to 3.645, *p* < 0.02), and on depersonalization/derealization symptoms (“*t*” test, t-values from 2.900 to 3.432 *p* < 0.005), and, at the last three evaluations, higher scores on post-traumatic stress (“*t*” test, t-values from 2.142 to 2.550, *p* < 0.05), with lower scores on health-related quality of life (“*t*” test, t-values from 3.079 to 3.375, *p* < 0.003).

Quality of Sleep. The frequency of patients with a score > 5 points on the Pittsburgh Sleep Quality Index gradually decreased, from 75.0% (95% C.I. 65.0–85.0%) at the first evaluation, to 69.1% (95% C.I. 58.0–80.0%) at the second evaluation, to 66.1% (95% C.I. 55.1–76.1%) at the third evaluation, and to 57.7% (46.3–69.1%) at the fourth evaluation. During the follow-up, at the four evaluations, patients with poor quality of sleep (score > 5) at the first evaluation had higher scores on the HADS total score (“*t*” test, t-values from 2.204 to 2.498, *p* < 0.04) and the anxiety subscore (“*t*” test, t-values from 2.257 to 3.744, *p* < 0.03), and on depersonalization/derealization at three evaluations (“*t*” test, t-values from 2.028 to 2.404, *p* < 0.05), as well as lower scores on health-related quality of life at two evaluations (“*t*” test, t = 2.007 and 2.515, *p* < 0.05). The Spearman correlation coefficients between the Sleep Quality Index at each evaluation with the other instruments administered in the same evaluation are described in Table 3, except for the MoCA score, which had no significant correlation. The most persistent strong relationships [39] were observed with the scores on the HADS, depersonalization/derealization symptoms, and health-related quality of life.

Anxiety and Depression. The frequency of patients with a HADS total score ≥ 11 decreased after the second evaluation, from 40.2% (95% C.I. 28.9–51.5%) at the first evaluation and 44.1% (95% C.I. 32.4–55.8%) at the second evaluation, to 29.5.1% (95% C.I. 19.0–40.1%) at the third evaluation, and 12.67% (4.9–20.3%) at the fourth evaluation. During the follow-up, patients who had a score ≥ 11 at the first evaluation, compared to those with a score < 11, had lower scores on health-related quality of life at the four evaluations (*t*” test, t-values from 4.538 to 4.703, *p* < 0.00003)

Depersonalization/derealization symptoms decreased over time. The ten most frequent symptoms are shown in Figure 3. Of note, at the first evaluation, the symptoms reported by more than half of the patients were “difficulty concentrating” (63%, 95% C.I. 51.9–74.1%), “difficulty focusing attention” (54%, 95% C.I. 43.5–64.5%), and “vision is dulled” (54%, 95%, C.I. 43.5–64.5%), which gradually decreased, while “difficulty understanding what others say to you” persisted in more than one third of the patients during the complete follow-up.

Dissociative experiences. During the follow-up, the frequency of a score > 8 on the Dissociative Experiences Scale decreased from 19.3% (95% C.I. 10.2–28.4%) at the first evaluation to 8.3% (95% C.I. 2.0–14.6%) at the fourth evaluation. During the follow-up, patients who had a score > 8 at the first evaluation had higher scores on sleep quality (worse sleep) (“*t*” test, t-values from 2.794 to 3.566, *p* < 0.007) and lower scores on health-related quality of life (“*t*” test, t-values from 2.583 to 3.469, *p* < 0.02) at the four evaluations, with higher scores on the HADS total score (“*t*” test, t-values from 2.591 to 3.318, *p* < 0.02), subscores (“*t*” test, t-values from 2.103 to 3.498, *p* < 0.04), and on post-traumatic stress (“*t*” test, t-values from 2.244 to 2.983, *p* < 0.03) at the last three evaluations, and on depersonalization/derealization symptoms (“*t*” test, t-values 2.237 and 4.040, *p* < 0.03) only at the first two evaluations. Comparisons of dyspnoea, pulse oximetry, and heart rate showed that patients with a score > 8 at the first evaluation also had the highest heart rates at the first 10 days and 11 to 50 days (“*t*” test, t-values 2.328 and 2.478, *p* < 0.04).

Post-traumatic stress disorder was diagnosed in 7 (9.7%, 95% C.I. 2.9–16.5%) patients at the first evaluation, in 3 (4.3%, 95% C.I. 0–8.9%) patients at the second evaluation, in 5 (6.3%, 95% C.I. 0.7–11.9%) patients at the third evaluation, and no patient fulfilled the criteria at the fourth evaluation. During the follow-up, patients with post-traumatic stress disorder had persistently higher scores on the HADS total score up to the third evaluation and anxiety subscore up to the fourth evaluation (“*t*” test, t-values from 2.047 to 4.935, *p* < 0.05), with higher scores on sleep quality (worse sleep) (“*t*” test, t-values from 2.372 to 3.539, *p* < 0.03) at the first, third, and fourth evaluations, with lower scores on health-related quality of life at the first, second, and fourth evaluations (“*t*” test, t-values from 2.615 to 3.634, *p* < 0.02), and a borderline significant difference at the second evaluation (“*t*” test, t = 1.980, *p* = 0.051). They also showed a lower MoCA score than those without the diagnosis, only at the first evaluation (21.1 ± 2.6 versus 24.9 ± 3.7, “*t*” test, t = 2.599, *p* = 0.011). Comparisons of the agenda records showed that patients with the diagnosis at the first evaluation had the highest heart rates during the 200-day follow-up (“*t*” test, t-values from 2.120 to 3.215 *p* < 0.04). They also had higher proportions of lymphocytes (“*t*” test, t = 2.420, *p* = 0.01) and lower proportions of neutrophils (“*t*” test, t = 3.531, *p* = 0.007) at hospital admission.

Cognitive performance. MoCA score < 25 was observed in 31 (43%, 95% C.I. 31.5–54.4%) patients at the first evaluation and in 23 (31.9%, 95% C.I. 21.1–42.6%) patients at all the following evaluations. However, significant score increases were evident only for the total score and the score on the language domain (Table 2). Patients with MoCA score < 25 at the first evaluation showed a higher HADS total score and anxiety subscore (“*t*” test, t-values 2.229 and 2.228, *p* < 0.03) and post-traumatic stress score (“*t*” test, t = 2.254, *p* < 0.03) only at the first evaluation. Comparisons of dyspnoea, pulse oximetry, and heart rate showed that patients with MoCA score < 25 at the first evaluation reported more dyspnoea during the follow-up than those with a score ≥ 25 during almost all the recordings (“*t*” test, t-values from 2.064 3 to 2.292, *p* < 0.03). Additionally, they showed the highest proportions of lymphocytes (“*t*” test, t = 2.039, p = 0.04) and lower proportions of neutrophils (“*t*” test, t = 2.629, *p* = 0.01) at hospital admission.

Health-related quality of life. During the follow-up, the total scores and subscores gradually increased except for the mental health domain (Table 2). However, during all the evaluations, the scores on general health perception and vitality were below 65%. At each of the four evaluations, the SF-36 total score was not related to patient age or the time since disease onset but with the scores on depersonalization/derealization, quality of sleep, and anxiety/depression (Spearman’s coefficients from 0.53 to 0.70, *p* < 0.00001), while moderate to strong correlations were evident with the score on dissociative experiences (Spearman’s coefficients from 0.43 to 0.57, *p* < 0.0002), and moderate correlation was found with the score on post-traumatic stress up to the third evaluation (Spearman’s coefficients from 0.23 to 0.50, *p* < 0.05).

### 3.2. Multivariate Analysis

#### 3.2.1. Cognitive Performance

Contribution to the variance in the MoCA score was evident from age, schooling, tobacco use, leukocyte count at hospital discharge, and HADS total score, while the repeated measures analysis (R in Table 4) showed that the increase in MoCA score was related to oxygen saturation at hospital admission, intubation for mechanical ventilation during hospitalization, and obesity (Table 4), with an interaction between intubation and obesity, since patients with obesity had higher scores when intubation was performed than when no intubation was performed, particularly at the first evaluation, when patients without obesity showed lower scores when intubation was performed (Figure 4). At each evaluation, these factors contributed to >25% of the variance in the MoCA score (Table 4), with an adjusted R^2^ of 0.40 (F = 4.861, *p* < 0.00002) at the first evaluation, 0.35 (F = 4.0191, *p* = 0.0001) at the second evaluation, 0.27 (F = 3.083, *p* = 0.002) at the third evaluation, and 0.29 (F = 3.332, *p* = 0.001) at the fourth evaluation. 

#### 3.2.2. Health-Related Quality of Life

At each evaluation, contribution to the variance in the SF-36 total score was evident from the first evaluation score on quality of sleep and depersonalization/derealization symptoms (Table 5). The repeated measures analysis (R in Table 5) showed that these covariates as well as the first 10 days dyspnoea score and gender contributed to the differences in the total score and the scores on physical problems, vitality, and emotional problems. At each evaluation, all of these variables contributed to at least one third of the variance of the total score (Table 5), with an adjusted R^2^ of 0.48 (F = 16.085, *p* < 0.00001) at the first evaluation, 0.39 (F = 12.126, *p* < 0.00001) at the second evaluation, 0.35 (F = 10.168, *p* < 0.00001) at the third evaluation, and 0.33 (F = 10.168, *p* < 0.00001) at the fourth evaluation. The most consistent contribution was from the report of depersonalization/derealization symptoms to the total score and to the scores on physical function, general health perception, emotional problems, and mental health dimensions. The dyspnoea score at the first 10 days contributed to the total score and to physical function, pain, and emotional problem dimensions, while gender; with lower scores in women compared to men up to the second evaluation, contributed to changes in the total score and to limitations due to physical problems, general health perception, vitality, and mental health dimensions (Table 5).

## 4. Discussion

The study was designed to assess correlations among physical signs, quality of sleep, common mental symptoms, and health-related quality of life after moderate to severe COVID-19 pneumonia. Daily agendas showed delayed decreases in dyspnoea scores compared to the records on pulse oximetry and heart rate, although changes in pulse oximetry were minimal during the follow-up. Apart from age and schooling, cognitive performance showed correlations mainly with the general clinical characteristics of the participants (obesity and tobacco use) and with the severity of disease (oxygen saturation at admission, intubation for mechanical ventilation during hospitalization, and leucocyte count at discharge), with an interaction between intubation and obesity. Scores on health-related quality of life were always related to scores on quality of sleep. Over months, health-related quality of life improved, while common mental symptoms decreased. The increase in health-related quality of life was related to decreases in depersonalization/derealization symptoms, which were mainly those related to attention and concentration, apart from the continuance of difficulty in understanding spoken language. The results support that the bidirectional relationship between quality of sleep and mental health may contribute to the persistence of mental symptoms and has an impact on health-related quality of life, with an opportunity for intervention.

The finding of delayed decrease in dyspnoea scores compared to pulse oximetry increase and heart rate decrease is consistent with the evidence that dyspnoea is not always related to cardiorespiratory function [40,41,42]. After one year of follow-up, even in patients with mild acute disease, respiratory symptoms may persist, with negative impact on employment, quality of life, and health care utilization [43], while the risk factor profile for the recovery of post-COVID-19 dyspnoea and for non-COVID-19 dyspnoea are similar [44]. In patients hospitalized with COVID-19, the main risk factors for persistent dyspnoea include older age, female sex, obesity, pre-existing mood disorders or cardiovascular disease, and long length of hospital stay [44]. The design of therapeutic strategies is still required, yet early exercise training rehabilitation may be of benefit [45].

Sleep and common mental symptoms. In the first evaluation of this study, 75% of the participants had poor sleep scores, with moderate decreases over time. During the follow-up, the quality of sleep score was related to symptoms of common mental disorders and contributed to the variability in health-related quality of life but had no correlation with cognitive performance. These findings are consistent with a six-month follow-up study on sleep disturbances showing frequencies >75% in 133 patients discharged from hospitalization due to COVID-19, with no influence from tobacco use, body mass index, gender, comorbidities, and length of hospital stay [46]. Of note, the participants of this study had moderate to severe disease, which may have increased the risk for poor sleep [47]. In adult populations, meta-analyses on the prevalence of sleep disturbances have shown geographical variation and higher prevalence in women than in men [47], with overall pooled prevalences of 28.9% (95% C.I. 25.7–32.3%) (*n* = 252,437) [47], 45% (95% C.I. 37–53%) (*n* = 15,362) [48], and 46% (95% C.I. 38–54%) (*n* = 13,935) [49]. On the other hand, a systematic review and meta-analysis of 18 follow-up studies on post-COVID-19 symptoms (*n* = 8591) showed that, one year after acute disease, the frequency of mental health and cognitive symptoms was 23% (95% C.I. 12–34%) for depression, 22%, (95% C.I. 15–29%) for anxiety, 19% (95% C.I. 7–31%) for memory loss, 18% (95% C.I. 2–35%) for concentration difficulties, and 12% (95% C.I. 7–17%) for insomnia [50].

The association between bad quality of sleep and mood disturbances, with no influence of oxygen saturation, is consistent with the results of deterministic whole-brain tractography on diffusion magnetic resonance imaging (MRI) scans showing disruptions in the limbic system after COVID-19, particularly the uncinate fasciculus, the cingulum cingulate, the cingulum hippocampus, and the arcuate fasciculus, with no significant differences between hospitalized and non-hospitalized patients [51]. Four weeks after acute COVID-19, psychiatric symptoms were evident in 56% of 402 adults (300 hospitalized), with correlation between the scores on depression and anxiety with systemic inflammation and no correlation with oxygen saturation [52]. Two and ten months after moderate to severe COVID-19, persistent poor sleep was related to increased inflammatory markers at two months and mental symptoms at ten months [16]. In rodents, among other effects, chronic sleep restriction induces enhancement of brain inflammatory molecules and attenuation of hippocampal brain-derived neurotrophic factor [53], pericyte detachment from capillary walls [54], and it interferes with blood–brain barrier function [55].

As could be expected, in this study, poor quality of sleep was related to post-traumatic stress disorder. However, it may have not interfered with the observed frequency of bad quality of sleep, since the diagnosis was <10% at the first evaluation and null at the fourth evaluation, supported by face-to-face psychiatric interviews, with a lower frequency over time if compared to previous reports [10].

Poor quality of sleep was also related to the report of depersonalization/derealization symptoms, which refers to the feeling of detachment from the self and from the surroundings. Evidence suggests that these symptoms may be mediated by circuits involved in the integration of sensory processing, the body schema, and the emotional experience [56]. They are frequent among the general population. In Germany, a prospective survey showed that 1.9% of 1287 participants (aged 14 to 90 years) had clinically significant symptoms, which were mainly associated with mood disorders, but also independently associated with chronic disease (i.e., hypertension, diabetes mellitus, chronic pulmonary disease), severe pain, and childhood adversities [57]. Meanwhile, evidence has shown a relationship between sleep disturbances and dissociative symptoms (for review see [58]). In this study, the inventory used to assess depersonalization/derealization symptoms was developed for clinically anxious patients [28].

The bidirectional relationship between quality of sleep, perception, and mental health could have contributed to the persistence of mental symptoms. The most frequent depersonalization/derealization symptoms were those resembling “brain fog,” with a gradual decrease. This finding is consistent with the results of a telephone survey of 400 adults at 6, 12, and 18 months after hospitalization due to COVID-19, showing decreases in the report of concentration loss and “self-perception of sluggish or fuzzy thinking” [59]. Nevertheless, the relationship between sleep disturbances and attention/concentration symptoms is consistent with the evidence by functional MRI showing that acute sleep deprivation alters brain functions related to attention tasks in the dorsolateral prefrontal cortex and the parietal sulcus [60], as well as the connection between the right precuneus and the right middle frontal gyrus, with differences before and after sleep deprivation in anxiety, attention, self-confidence, anger, and nervousness [61]. After 6 months post-hospitalization due to COVID-19, compared with healthy subjects, patients may have a higher amplitude of low-frequency fluctuations in the right precuneus, middle temporal gyrus, and middle and inferior occipital gyrus, and lower amplitude in the right middle frontal gyrus and bilateral inferior temporal gyrus, with increased functional connectivity between the right middle occipital gyrus and the left inferior occipital gyrus, and reduced functional connectivity between the right inferior occipital gyrus and right inferior temporal gyrus/bilateral fusiform gyrus and between the right middle frontal gyrus and right middle frontal gyrus/supplementary motor cortex/precuneus [62].

The sustained report of “difficulty understanding what others say to you” by circa one third of the participants of this study is consistent with the clinical evidence that, after acute COVID-19, some patients may have difficulty in discriminating subtle temporal aspects of sound signals [63], with other auditory processing deficits [64,65]. Two auditory pathways have been considered for speech perception, one for temporal information from the cerebellum to the frontal cortex via the thalamus, and one for retrieval of representations in the temporal cortex, which are projected to the frontal cortex [66]. Then, deficits could be related to the known sensitivity of cerebellar and medial temporal lobe structures to hypoxic injury [67]. A systematic review of 90 neuroimaging and neuropathological studies showed consistent impairment of white matter, brainstem, and frontotemporal areas [68]. An overview of studies examining the cerebral microstructure and function related to COVID-19 showed changes in the cerebral microstructure mainly allocated in the insula, superior temporal gyrus, hippocampus, and thalamic radiation tract [69]. However, intended functional studies are required to evaluate impairment/recovery of central auditory processing after COVID-19.

Cognitive performance. In this study, no differences in the frequency of MoCA score < 25 were observed after the second evaluation. A systematic review and metanalysis of 74 follow-up studies showed similar results, with no statistically significant difference in the report of cognitive impairment at <6 and ≥6 months after acute disease or between inpatients and outpatients [1].

Additionally, in this study, changes in cognitive performance during the follow-up were mainly related to variables associated with the acute phase of the disease. In agreement with a web-optimized assessment (*n* = 81,337), participants who were infected by SARS-CoV-2 displayed persistent cognitive deficits, after controlling for premorbid intelligence, pre-existing comorbidities, socio-demographic factors, and mental health symptoms [70]. This evidence supports the influence of hypoxia and inflammation with hypometabolism [71], with acute inflammation contributing to white matter abnormalities [72,73].

The results of this study also showed an interaction between obesity and intubation for mechanical ventilation, in which higher MoCA scores were observed in patients with obesity versus without invasive mechanical ventilation. In a post-COVID-19 clinic (*n* = 321; 107 outpatients/214 inpatients), cognitive performance was independent of severe disease or markers of physical, cognitive, and pulmonary health [74]. Although, prolonged mechanical ventilation due to severe COVID-19 is related to declines in physical and mental health [75]. After COVID-19 pneumonia, patients receiving invasive respiratory assistance in the acute phase can show better cognitive functions in the subacute phase than those with ventilatory masks, according to their clinical context [76]. Nevertheless, it is worth bearing in mind the evidence suggesting that, three months after mild/moderate COVID-19, screening tests may not reliably detect cognitive dysfunction [77].

Quality of life. In this study, increasing total score and subscores on health-related quality of life during a >2-year follow-up was mainly due to improvements in physical function and emotional problems (Table 5). This result was consistent with the objective improvements observed within the daily agendas and the gradual decreases in the scores on anxiety/depression and chronic stress. Evidence supports that fear and anxiety may lower the level of self-efficacy and increase distress, suggesting the promotion of psycho-educational interventions for adoption of appraisal strategies that could be beneficial for perceived self-efficacy in COVID-19 survivors [78]. However, the general health perception and vitality subscores were low at the four evaluations (Table 2), with slight changes related to decreases in depersonalization/derealization symptoms for the two subscores and to improvements in sleep quality scores for the vitality subscore. In addition, at each evaluation, simple correlation analysis showed that both the total score and all subscores on health-related quality of life were related to sleep quality scores. Of note, the main depersonalization/derealization symptoms were those related to attention/concentration deficits, with a negative impact on perceived health-related quality of life, which persisted after >2 years of follow-up.

The repercussions of COVID-19 are complex, with significant impact on mental health that may diminish health-related quality of life. Two years after COVID-19, physical and mental health may improve, while a variety of sequelae can remain and require further investigation. In addition to physical signs and emotional symptoms, bad quality of sleep and attention/concentration deficits may contribute to the general health perception. Early assessment of quality of sleep and depersonalization/derealization symptoms may contribute to designing tailored intervention to facilitate recovery. Repeated functional imaging studies, accompanied by assessments of mental health (including perception) and functioning, could be useful to better understand COVID-19 sequelae. The information of this and further studies could be useful to plan mental support services during rehabilitation after acute COVID-19 and to identify clinical research priorities.

The main limitations of this study are the single location, which allowed standardized treatment during the acute phase; the small sample size, after selective recruitment to avoid the main potential confounders; the lack of a control group of inpatients without COVID-19 due to hospitalization constraints during the study period; and the lack of mental symptom ratings before or during the acute phase of COVID-19. However, data collection was standardized, and the study protocol was rigorously applied during the follow-up, in which assessments were always performed in the same setting and by the same investigator, with face-to-face psychiatric interviews, to minimize subjectivity.

## 5. Conclusions

Multiple factors may contribute to the mental symptoms related to post-COVID-19 conditions, including physical and mental health, as well as idiosyncrasies. Apart from the respiratory symptoms at hospital discharge and gender, quality of sleep and attention/concentration deficits contribute to health-related quality of life, offering opportunities for both early detection and intervention to promote recovery. After acute COVID-19, mental support services, including sleep hygiene and personalized sleep treatment, could be beneficial both at hospital discharge and during rehabilitation.

## Figures and Tables

**Figure 2 biomedicines-12-01574-f002:**
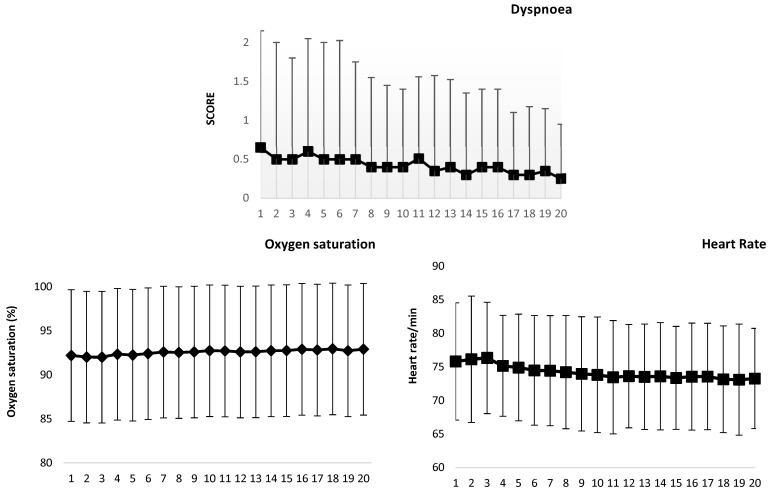
Median and quartiles 1 and 3 of the dyspnoea score and mean and standard deviation of the mean pulse oximetry and heart rate of 71 patients, by 20 sets of 10 days each, for 200 days.

**Figure 3 biomedicines-12-01574-f003:**
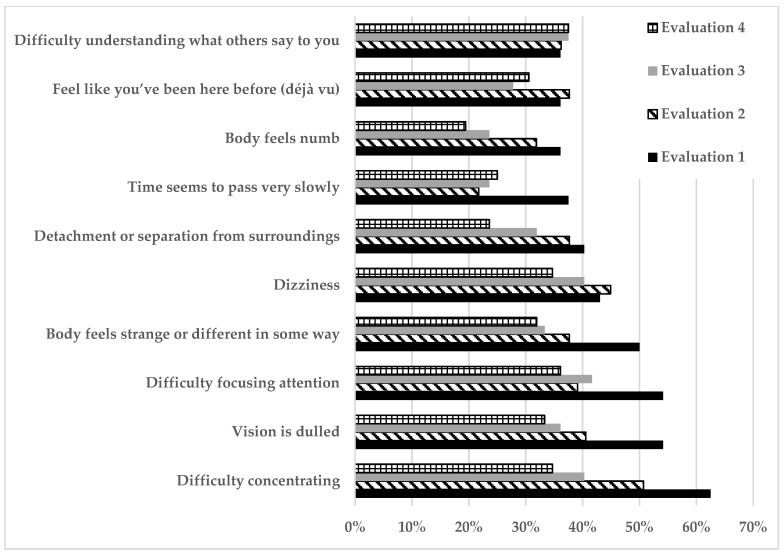
The 10 most frequent depersonalization/derealization symptoms at the four evaluations.

**Figure 4 biomedicines-12-01574-f004:**
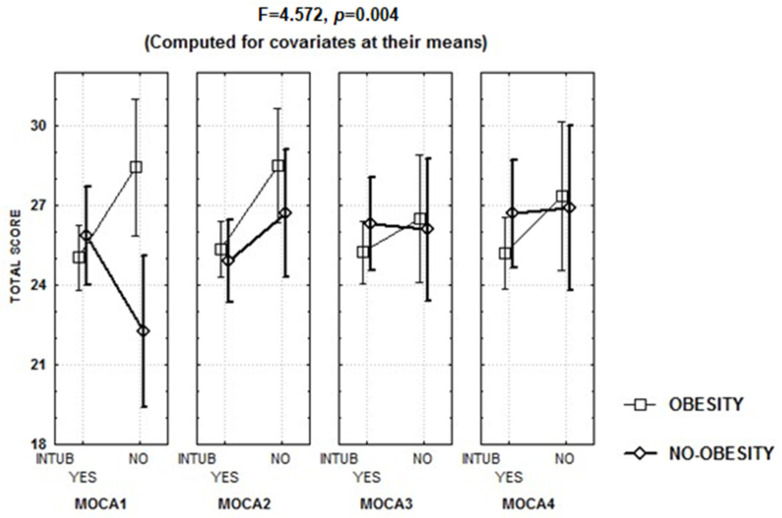
Mean and standard error of the mean MoCA scores during the four evaluations, according to obesity and the need of intubation for mechanical ventilation during hospitalization (INTUB).

**Table 2 biomedicines-12-01574-t002:** Summary of the scores of the 72 patients participating in the study at the four evaluations. The result of the Friedman test on each instrument is provided, including the statistical significance and the *X*^2^ value (*p*(*X*^2^ value)).

Assessment Questionnaires	Evaluation 1 (*n* = 72)	Evaluation 2 (*n* = 69)	Evaluation 3 (*n* = 72)	Evaluation 4 (*n* = 72)	
Days Since Start of Disease	143 (69)	241(66)	339 (75)	889 (178)	
	Mean (Range)	Mean (Range)	Mean (Range)	Mean (Range)	*p* (*X*^2^ Value)
Sleep index	8.1 (1–19)	7.9 (0–19)	7.5 (1–19)	7.0 (1–15)	0.002 (14.37)
HADS total score	10.2 (0–29)	9.3 (0–28)	7.8/0–28)	6.0 (0–20)	<0.0001 (16.291)
Anxiety subscale	6.0 (0–18)	5.6 (0–16)	4.3 (0–19)	3.2 (0–15)	<0.0001 (46.54)
Depression subscale	4.1 (0–17)	3.7 (0–14)	3.4 (0–15)	2.7 (0–9)	0.01 (10.15)
Depersonalization/derealization	10 (0–77)	7.6 (0–59)	7 (0–49)	4 (2–7)	0.0006 (17.06)
Dissociative experience scale	5.7 (0–50)	4.1 (0–19.6)	3.5 (0–32.5)	3.3 (0–28)	<0.00001(31.79)
Post-traumatic stress	5.4 (0–45)	3.4 (0–41)	3.4 (0–52)	0.66 (0–19)	0.00002 (8.42)
Health-related quality of life (SF-36)					
Physical function	66 (20–100)	73 (20–100)	75 (15–100)	80 (45–100)	<0.00001(56.98)
Limitations physical problems	50 (0–100)	64 (0–100)	65 (0–100)	75 (0–100)	<0.00001 (43.23)
Pain	58 (10–100)	69 (22–100)	70 (10–100)	82 (31–100)	<0.00001 (61.24)
General health perception	59 (5–100)	59 (5–100)	63 (10–100)	63 (10–97)	0.005(12.60)
Vitality	52 (5–95)	59 (5–100)	58 (0–100)	61 (10–100)	0.0003 (18.72)
Social functioning	67 (0–100)	75 (0–100)	74 (0–100)	82 (50–100)	<0.00001(37.63)
Limitations emotional problems	63 (0–100)	68 (0–100)	72 (0–100)	82 (33–100)	<0.00001 (40.92)
Mental health	73 (16–100)	77 (36–100)	75 (4–100)	76 (36–100)	0.298 (3.67)
SF-36 total score	62 (17–95)	68 (24–97)	69 (12–100)	75 (38–98)	<0.00001 (69.42)
MoCA					
Visuo-spatial	3.9 (0–5)	4.1 (0–5)	5 (0–5)	4 (0–5)	0.092 (6.41)
Naming	2.8 (0–3)	2.9 (2–3)	2.9 (1–3)	2.9 (2–3)	0.691 (1.45)
Attention	4.6 (1–6)	4.4 (1–6)	4.4 (1–6)	4.6 (1–6)	0.130 (5.64)
Language	2.3 (0–3)	2.2 (0–3)	2.3 (1–3)	2.6 (1–3)	<0.00001 (29.69)
Abstraction	1.3 (0–2)	1.4 (0–2)	1.4 (0–2)	1.3 (0–2)	0.582 (1.95)
Memory	3.4 (0–5)	3.7 (0–5)	3.7 (0–5)	3.7 (1–5)	0.140 (5.46)
Orientation	5.9 (4–6)	5.9 (5–6)	5.9 (4–6)	5.9 (5–6)	0.491 (2.40)
MoCA total score	24.5 (13–30)	24.9 (14–30)	25.1 (16.7–30)	25.5 (14–30)	0.002 (14.64)

**Table 3 biomedicines-12-01574-t003:** Spearman correlation between the score on the Pittsburgh Sleep Quality Index and other instruments within each evaluation, including the number of participants, the t-value, and the statistical significance (*p*).

Assessment Questionnaires	Evaluation 1 (*n* = 72)	Evaluation 2 (*n* = 69)	Evaluation 3 (*n* = 72)	Evaluation 4 (*n* = 72)
	Spearman’s R (t-Value, *p*)	Spearman’s R (t-Value, *p*)	Spearman’s R (t-Value, *p*)	Spearman’s R (t-Value, *p*)
HADS total score	0.58 (6.100, <0.00001)	0.57 (5.701, <0.00001)	0.53 (5.229, <0.00001)	0.58 (6.013, 0.00001)
Anxiety subscale	0.67 (7.572, <0.00001)	0.57 (5.736, <0.00001)	0.52 (5.216, <0.00001)	0.55 (5.511, <0.00001)
Depression subscale	0.35 (3.146, 0.002)	0.47 (4.380, 0.00004)	0.49 (4.759, 0.00001)	0.47 (4.494, 0.00002)
Depersonalization/ Derealization	0.55 (5.543, <0.00001)	0.55 (5.450, <0.00001)	0.50 (4.916, <0.00001)	0.40 (3.673, 004)
Dissociative experience scale	0.43 (4.035, 0.0001)	0.28 (2.415, 0.018)	0.36 (3.238, 0.001)	0.25 (2.227, 0.029)
Post traumatic stress scale	0.42 (3.954, 0.0001)	0.32 (2.780, 0.007)	0.41 (3.854, 0.0002)	−
SF36 total score	−0.53 (−5.336, <0.00001)	−0.68 (−7.597, <0.00001)	−0.67 (−7.731, <0.00001)	−0.67 (−7.676, <0.00001)
Physical function	−0.41 (−3.807, 0.0002)	−0.47 (−4.419, 0.00003)	−0.45 (−4.310, 0.00005)	−0.48 (−4.587, 0.00001)
Limitations physical problems	−0.41 (−3.807,0.0002)	−0.60 (−6.153, <0.00001)	−0.56 (−5.689, <0.00001)	−0.53 (−5.346, <0.00001)
Pain	−0.48 (−4.647, 0.00001)	−0.55 (−5.421, <0.00001)	−0.57 (−5.830, <0.00001)	−0.56 (−5.698, <0.00001)
General health perception	−0.27 (−2.431, 0.017)	−0.47 (−4.402, 0.00003)	−0.48 (−4.636, 0.00001)	−0.37 (−3.403, 0.001)
Vitality	−0.46 (−4.366, 0.00004)	−0.55 (−5.438, <0.00001)	−0.59 (−6.272, <0.00001)	−0.66 (−7.417, <0.00001)
Social functioning	−0.32 (−2.893, 0.005)	−0.56 (−5.664, <0.00001)	−0.60 (−6.420, <0.00001)	−0.50 (−4.949, <0.00001)
Limitations emotional problems	−0.36 (−3.310, 0.001)	−0.47 (−4.386, 0.00004)	−0.58 (−6.029, <0.00001)	−0.50 (−4.914, <0.00001)
Mental health	−0.51 (−5.025, <0.00001)	−0.56 (−5.641, <0.00001)	−0.57 (−5.846, <0.00001)	−0.57 (−5.885, <0.00001)

**Table 4 biomedicines-12-01574-t004:** Multivariate analysis of covariance on the total score of the Montreal Cognitive Assessment (MoCA) of the 69 patients who completed the four evaluations. The statistical significance (*p*) and the F value are described for the total scores and for the repeated measures of the total scores (R).

Variable	*p* (F)
Intercept	<0.00001 (86.699)
Age	0.030(4.952)
Schooling	0.004(8.785)
Oxygen saturation at hospital admission	0.215 (1.567)
HADS total score	0.001 (10.927)
Leukocyte count at hospital discharge	0.006 (8.079)
Tobacco use	0.011 (6.846)
Obesity	0.441 (0.601)
Intubation	0.266 (1.259)
Tobacco use × Obesity	0.404 (0.704)
Tobacco use × Intubation	0.662 (0.192)
Obesity × Intubation	0.121 (2.471)
Tobacco use × Obesity × Intubation	0.608 (0.265)
Repeated Measures (R)	0.001 (5.569)
R × Age	0.149 (1.796)
R × Schooling	0.720 (0.445)
R × Oxygen saturation at hospital admission	0.0005 (6.186)
R × HADS total score	0.60 (2.515)
R × Leukocyte count at hospital discharge	0.124 (1.945)
R × Tobacco use	0.096 (2.141)
R × Obesity	0.001 (5.165)
R × Intubation	0.026 (3.162)
R × Tobacco use × Obesity	0.286 (1.268)
R × Tobacco use × Intubation	0.455 (0.875)
R × Obesity × Intubation	0.004 (4.572)
R × Tobacco use × Obesity × Intubation	0.284 (1.275)

**Table 5 biomedicines-12-01574-t005:** Multivariate analysis of covariance in the health-related quality of life scores (total and by dimension) on the results of the 69 patients who completed the four evaluations, including the statistical significance (*p*) and the F value.

Variable	TOTAL Score	Physical Function	Physical Problems	Pain	General Health	Vitality	Social Functioning	Emotional Problems	Mental Health
	*p* (F)	*p* (F)	*p* (F)	*p* (F)	*p* (F)	*p* (F)	*p* (F)	*p* (F)	*p* (F)
Intercept	<0.00001 (413.98)	<0.00001 (303.40)	<0.00001 (102.45)	<0.00001 (276.22)	<0.00001 (194.70)	<0.00001 (242.08)	<0.00001 (423.52)	<0.00001 (166.12)	<0.00001 (682.67)
First Pittsburgh Sleep Quality Index score	0.009 (7.236)	0.072 (3.3460)	0.035 (4.610)	0.013 (6.438)	0.453 (0.569)	0.020 (5.632)	0.020 (5.676)	0.200 (1.674)	0.00025 (15.055)
First Depersonalization/ Derealization score	0.0002 (15.519)	0.152 (2.097)	0.063 (3.566)	0.032 (4.763)	0.0005 (13.286)	0.004 (8.803)	0.002 (9.939)	<0.00001 (28.300)	0.001 (10.780)
First 10 days median dyspnea	0.756 (0.096)	0.463 (0.544)	0.478 (0.508)	0.388 (0.754)	0.823 (0.050)	0.219 (1.537)	0.137 (2.263)	0.812 (0.056)	0.017 (6.003)
Gender	0.225 (1.499)	0.0006 (7.804)	0.287 (1.153)	0.173 (1.898)	0.125 (2.411)	0.166 (1.963)	0.288 (1.147)	0.933 (0.007)	0.492 (0.477)
Repeated measures (R)	0.004 (4.511)	0.208 (1.526)	0.034 (2.928)	0.886 (0.214)	0.228 (1.455)	0.0008 (5.774)	0.305 (1.214)	0.040 (2.819)	0.428 (0.928)
R × Pittsburgh Sleep Quality Index score	0.053 (2.599)	0.408 (0.968)	0.150 (1.789)	0.367 (1.060)	0.62 (2.482)	0.001 (5.295)	0.573 (0.667)	0.286 (1.268)	0.142 (1.836)
R × Depersonalization/ Derealization score	0.0001 (7.116)	0.001 (5.119)	0.187 (1.613)	0.108 (2.047)	0.016 (3.513)	0.006 (4.258)	0.279 (1.289)	0.002 (4.851)	0.005 (4.363)
R × First 10 days dyspnoea score	0.00009 (7.455)	0.0001 (7.094)	0.113 (2.010)	0.00009 (7.450)	0.647 (0.552)	0.247 (1.389)	0.075 (2.335)	0.0006 (5.994)	0.620 (0.593)
R × Gender	0.002 (5.091)	0.289 (1.261)	0.045 (2.717)	0.885 (0.215)	0.004 (4.453)	0.004 (4.493)	0.786 (0.353)	0.247 (1.401)	0.001 (5.615)

## Data Availability

All data generated or analyzed during this study are included in this article. Further enquiries can be directed to the corresponding author.

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
