# Peer review of "Quality of Sleep and Mental Symptoms Contribute to Health-Related Quality of Life after COVID-19 Pneumonia, a Follow-Up Study of More than 2 Years"

_biomedicines, 2024, doi:10.3390/biomedicines12071574_

Round 1

Reviewer 1 Report

Comments and Suggestions for Authors

Here is the detailed review of the provided article, highlighting its strengths and pinpointing areas that need improvement, line by line.

Strengths:

  1. Title and Authors:

    • The title is clear and descriptive, indicating the study's focus on sleep quality, mental symptoms, and quality of life post-COVID-19.
    • The authors are affiliated with recognized institutions, lending credibility to the research.
  2. Abstract:

    • The abstract is concise and effectively summarizes the study's objectives, methods, results, and conclusions.
    • The structure is well-organized, facilitating the understanding of key points.
  3. Introduction:

    • The introduction provides a global context on the issue of post-COVID-19 symptoms, using recent and relevant data.
    • The citations are well-integrated to support the claims made.
  4. Methods:

    • The methods are described in detail, allowing for the study's replicability.
    • The inclusion of various assessment tools (e.g., Pittsburgh Sleep Quality Index, HADS) adds rigor to the research.
  5. Results:

    • The results are presented with tables and figures that facilitate data comprehension.
    • The statistical analysis is detailed and appropriate for the type of data collected.
  6. Discussion:

    • The discussion interprets the results comprehensively, linking them to existing literature.
    • The study's limitations are acknowledged, demonstrating transparency.

Areas for Improvement:

  1. Line 2-3: "Quality of sleep and mental symptoms contribute to health-re- 2 lated quality of life after COVID-19 pneumonia, a >2 years fol- 3 low-up study"

    • "a >2 years follow-up study" should be "a follow-up study of more than 2 years."
  2. Line 17: "assess correlations among physical signs, quality of 16 sleep, common mental symptoms, and health related quality of life"

    • "health related quality of life" should be "health-related quality of life."
  3. Line 18: "Daily changes on dyspnoea and pulse oximetry were recorded (200 days)"

    • "on dyspnoea" should be "in dyspnoea."
  4. Line 23: "though, changes on pulse oximetry 23 were minimal"

    • "though," should be removed or replaced with "however."
  5. Line 27: "during recovery, above all the study variables, bad quality of sleep and mental symptoms"

    • "above all the study variables" sounds redundant and could be rephrased.
  6. Line 34-35: "The World Health Organization refers to post-COVID-19 conditions as “long-term 34 symptoms that some people experience after they have had COVID-19” [1]. Globally, ac- 35 cording to a conservative estimated incidence of 10% of infected people [2], at least 77"

    • The WHO citation should be more smoothly integrated into the text.
  7. Figure 1 and Tables:

    • The tables and figures are useful but could benefit from more detailed captions to better contextualize the data.
  8. Line 84: "After approval by the institutional Research and Ethics Committees"

    • Should include a clearer description of the ethical approval process.
  9. Line 87: "Seventy-two patients (aged 25-to 85 years, 30 men/42 women) gave written informed 88 consent to participate in the study"

    • It could be helpful to explain in detail how informed consent was obtained.
  10. Line 113: "Table 1. General characteristics of the 72 patients participating in the study by gender."

    • The data presented in the tables should include additional statistical information, such as confidence intervals, to enhance comprehensibility.
  11. Discussion Section:

    • Some parts of the discussion could be further developed to provide a more comprehensive view of the implications of the results.
  12. Conclusions:

    • The conclusions could be strengthened with more concrete suggestions for future research or clinical applications.

The article is well-structured and significantly contributes to the understanding of the long-term effects of COVID-19 on sleep quality and mental symptoms. However, stylistic improvements and some additional details could further increase the clarity and impact of the study.

The authors are advised to cite the following article to strengthen their study and provide a broader context on the psychological effects of COVID-19:

Diotaiuti, P., Valente, G., Mancone, S., Corrado, S., Bellizzi, F., Falese, L., Langiano, E., Vilarino, G. T., & Andrade, A. (2023). Effects of Cognitive Appraisals on Perceived Self-Efficacy and Distress during the COVID-19 Lockdown: An Empirical Analysis Based on Structural Equation Modeling. International journal of environmental research and public health20(7), 5294. https://doi.org/10.3390/ijerph20075294

Author Response

We thank the reviewer for the positive comments and for all the recommendations, editions were made accordingly, text editions are underlined.

Comment 1: Line 2-3: "Quality of sleep and mental symptoms contribute to health-related quality of life after COVID-19 pneumonia, a >2 years follow-up study"

    • "a >2 years follow-up study" should be "a follow-up study of more than 2 years."

The sentence has been edited as suggested (lines 2-3).

Comment 2: Line 17: "assess correlations among physical signs, quality of 16 sleep, common mental symptoms, and health related quality of life"

    • "health related quality of life" should be "health-related quality of life."

We thank the reviewer for the observation. The hyphen has been introduced between the words, and the manuscript was revised.

Comment 3: Line 18: "Daily changes on dyspnoea and pulse oximetry were recorded (200 days)"

    • "on dyspnoea" should be "in dyspnoea."

We thank the reviewer for the observation. The preposition has been changed (line 18).

Comment 4: Line 23: "though, changes on pulse oximetry 23 were minimal"

    • "though," should be removed or replaced with "however."

The word however was introduced (line 23).

Comment 5: Line 27: "during recovery, above all the study variables, bad quality of sleep and mental symptoms"

    • "above all the study variables" sounds redundant and could be rephrased.

The sentence has been rephrased (lines 27-29).

Comment 6: Line 34-35: "The World Health Organization refers to post-COVID-19 conditions as “long-term 34 symptoms that some people experience after they have had COVID-19” [1]. Globally, ac- 35 cording to a conservative estimated incidence of 10% of infected people [2], at least 77"

    • The WHO citation should be more smoothly integrated into the text.

The paragraph has been edited accordingly (lines 34-37).

Comment 7: Figure 1 and Tables:

    • The tables and figures are useful but could benefit from more detailed captions to better contextualize the data.

The captions were edited to add information.

Comment 8: Line 84: "After approval by the institutional Research and Ethics Committees"

    • Should include a clearer description of the ethical approval process.

The description was incorporated (lines 87-89).

Comment 9:  Line 87: "Seventy-two patients (aged 25-to 85 years, 30 men/42 women) gave written informed 88 consent to participate in the study"

    • It could be helpful to explain in detail how informed consent was obtained.

The explanation is now provided (lines 98-109).

Comment 10: Line 113: "Table 1. General characteristics of the 72 patients participating in the study by gender."

    • The data presented in the tables should include additional statistical information, such as confidence intervals, to enhance comprehensibility.

Tables were edited to add 95% C.I. or ranges as appropriate.

Comment 11: Discussion Section:

    • Some parts of the discussion could be further developed to provide a more comprehensive view of the implications of the results.

The Discussion has been edited cautiously to prevent that excessive information could preclude interest for a wide audience. 

Comment 12: Conclusions:

    • The conclusions could be strengthened with more concrete suggestions for future research or clinical applications.

The final paragraph of the discussion (before ethe limitations) was edited to suggest future research (lines 523-533) and a Conclusions section was added at the end of the manuscript (lines 543-549).  

Comment 13: The article is well-structured and significantly contributes to the understanding of the long-term effects of COVID-19 on sleep quality and mental symptoms. However, stylistic improvements and some additional details could further increase the clarity and impact of the study.

We thank the reviewer for the positive comments and excellent recommendations.

Comment 14: The authors are advised to cite the following article to strengthen their study and provide a broader context on the psychological effects of COVID-19: Diotaiuti, P., Valente, G., Mancone, S., Corrado, S., Bellizzi, F., Falese, L., Langiano, E., Vilarino, G. T., & Andrade, A. (2023). Effects of Cognitive Appraisals on Perceived Self-Efficacy and Distress during the COVID-19 Lockdown: An Empirical Analysis Based on Structural Equation Modeling. International journal of environmental research and public health20(7), 5294. https://doi.org/10.3390/ijerph20075294

The reference has been included in the Discussion (lines 513-514).

Reviewer 2 Report

Comments and Suggestions for Authors

The topic of the manuscript entitled “Quality of sleep and mental symptoms contribute to health-related quality of life after COVID-19 pneumonia, a >2 years follow-up study” is very interesting. A strength of the manuscript is that it is a longitudinal study and that multiple variables were measured. Despite this interest, the authors should make an effort to present the information more clearly. This would make the manuscript easier to understand.

The language is used imprecisely, which generates confusion. Some examples are given below:

-In the abstract, on page 1, lines 26 to 28 it says "during recovery, above all the study variables, bad quality of sleep and mental symptoms (mainly attention/concentration) contributed to health perception and vitality on the health-related quality of life scores". It is specified that it is about the bad quality of sleep but the meaning of perception and vitality is not clear.

 -On Page 5, lines 164, 165 it states: “The Montreal Cognitive Assessment (MoCA) [29] that comprises six domains: memory, executive functioning, attention, language, visuospatial, and orientation”. But these domains do not coincide exactly with those in Table 2 (page 7) where the following are listed: Visuo-spatial, Naming, Attention, Language, Abstraction, Memory, Orientation

-Table 2 (page 7), in the Health related quality of life (SF-36) it says "Emotional problems", and "physical problems" but Table 3 it says "Limitations emotional problems" and "Limitations physical problems".

-On page 8, lines 233 and 234 it says "Table 3. Spearman correlation between the score on the Pittsburgh Sleep Quality Index and other instruments within each evaluation". But the correlation presented is with other variables, not with other instruments. In addition to the fact that it is more accurate to use the term "variable," it should be noted that some of the instruments used assess more than one variable.

-On page 2, lines 79 to 82 it states that the second aim of the study is "To explore correlations among quality of sleep, psychological symptoms, cognitive performance, and health related quality of life at four time points, in a >2 years follow-up of adults surviving moderate to severe COVID-19 pneumonia, taking into account their general characteristics”. It should be clearly explained what "general characteristics" refers to.

-Page 5, lines 161 to 163 describes Posttraumatic Stress Disorder Symptom Severity Scale but does not give the cut-off points for establishing the diagnostic criteria.

-In Table 2 (page 7) the Post traumatic stress disorder scores are not shown and the reason for this is not explained.

-The Statistical Analysis section should be revised and the statistical analyses performed should be clearly and completely explained as it is described in a very confusing way. Furthermore, in this section (on line 179) it says "Bivariate analysis included "t" test for independent samples" but in the study presented the samples are not independent, since they are the same patients.

-In multivariate covariance analyses, it should be clearly explained which variables were considered as independent and which were considered as covariates.

- Data in the text should be revised and presented more clearly, preferably in tables.

Author Response

Comment 1: The topic of the manuscript entitled “Quality of sleep and mental symptoms contribute to health-related quality of life after COVID-19 pneumonia, a >2 years follow-up study” is very interesting. A strength of the manuscript is that it is a longitudinal study and that multiple variables were measured. Despite this interest, the authors should make an effort to present the information more clearly. This would make the manuscript easier to understand.

We thank the reviewer for the positive comments and the useful observations and recommendations, Editions are underlined.

Comment 2:  The language is used imprecisely, which generates confusion. Some examples are given below:- In the abstract, on page 1, lines 26 to 28 it says "during recovery, above all the study variables, bad quality of sleep and mental symptoms (mainly attention/concentration) contributed to health perception and vitality on the health-related quality of life scores". It is specified that it is about the bad quality of sleep but the meaning of perception and vitality is not clear.

Thank you for the observation, the sentence has been edited for clarity (lines 27-29).

Comment 3: On Page 5, lines 164, 165 it states: “The Montreal Cognitive Assessment (MoCA) [29] that comprises six domains: memory, executive functioning, attention, language, visuospatial, and orientation”. But these domains do not coincide exactly with those in Table 2 (page 7) where the following are listed: Visuo-spatial, Naming, Attention, Language, Abstraction, Memory, Orientation

Thank you for the observation, the description in the text has been edited to be identical to the description in the Table (lines 179-180).

Comment 4: Table 2 (page 7), in the Health related quality of life (SF-36) it says "Emotional problems", and "physical problems" but Table 3 it says "Limitations emotional problems" and "Limitations physical problems".

Thank you for the observation, the description is now identical in the Two Tables.

Comment 5: On page 8, lines 233 and 234 it says "Table 3. Spearman correlation between the score on the Pittsburgh Sleep Quality Index and other instruments within each evaluation". But the correlation presented is with other variables, not with other instruments. In addition to the fact that it is more accurate to use the term "variable," it should be noted that some of the instruments used assess more than one variable.

Thank you for the observation, the label has been changed to read “Assessment Questionnaires”.

Comment 6: On page 2, lines 79 to 82 it states that the second aim of the study is "To explore correlations among quality of sleep, psychological symptoms, cognitive performance, and health related quality of life at four time points, in a >2 years follow-up of adults surviving moderate to severe COVID-19 pneumonia, taking into account their general characteristics”. It should be clearly explained what "general characteristics" refers to.

Thank you for the recommendation, the sentence has been edited to specify “the general clinical characteristics” (line 85).

Comment 7: Page 5, lines 161 to 163 describes Posttraumatic Stress Disorder Symptom Severity Scale but does not give the cut-off points for establishing the diagnostic criteria.

Thank you for the comment. Individual diagnosis was performed according to both the instrument and face-to-face psychiatric interview (lines 176-177).

Comment 8: In Table 2 (page 7) the Post traumatic stress disorder scores are not shown and the reason for this is not explained.

The scores are now included in Table 2. Although, individual diagnosis was performed according to both the instrument and face-to-face psychiatric interview (lines 176-177).

Comment 9: The Statistical Analysis section should be revised and the statistical analyses performed should be clearly and completely explained as it is described in a very confusing way. Furthermore, in this section (on line 179) it says "Bivariate analysis included "t" test for independent samples" but in the study presented the samples are not independent, since they are the same patients.

Thank you for the recommendation. The Section has been edited to describe the use of the tests (lines 192-201).

Comment 10: In multivariate covariance analyses, it should be clearly explained which variables were considered as independent and which were considered as covariates.

Thank you for the recommendation. The Section was edited to identify the outcome variables (lines 199-201) , while the covariates are described on Tables 4 and 5

Comment 11:  Data in the text should be revised and presented more clearly, preferably in tables.

Thank you for the recommendation. The description was revised, it provides the required information to support each statement, while more detailed information is provided in Tables 1 to 5.

Reviewer 3 Report

Comments and Suggestions for Authors

ABSTRACT

Please, add mean age of the sample.

INTRODUCTION

More information on how post-covid influenced sleep quality is needed.

Similarly, the author should describe in more detail what are “affective” symptoms. Are they referring to psychological symptoms?

May be the authors should consider referring to “mental health” instead of “psychological symptoms”.

METHODS

The participants “flow” is not clear. How many participants were invited to take part in the research and how many of them were initially recruited? How many fulfilled inclusion criteria?

Figure 1 “says” an initial sample of 79, while in the text it seems that is 74.

The characteristics (model, reliability and validity) of the commercial bands used to assess pulse oximetry and heart rate

Were the patients interviewed individually? By whom?

RESULTS

Please, provide information on how many patients completed all the assessments. Were all the patients able to perform valid self-evaluations every day during 200 days?

Were age or sex considered as cofounding variables?

DISCUSSION

My advice is to begin by indicating what was the main aim of the research and what were the main findings. Also, what is the utility of the provided results.

What are the factors that may affect sleep quality during the whole period?

The final paragraph shows synthetized information on the evolution of the variables assessed during the whole period. However, it is not clear enough. I encourage the authors more clearly what was the evolution on quality of sleep, mental health and quality of life observed in the participants and whether an improvement or worsening in a specific variable led to changes in other variables.

I think that a separate and independent “Conclusion” subsection would be welcomed.

Author Response

We thank the reviewer for all the valuable recommendations.

ABSTRACT

Comment 1: Please, add mean age of the sample.

Thank you for the observation, the mean age has been included in the Abstract (line 21).

INTRODUCTION

Comment 2: More information on how post-covid influenced sleep quality is needed.

A paragraph has been added to the introduction with reference to a review on the topic (lines 71-77).

Comment 3: Similarly, the author should describe in more detail what are “affective” symptoms. Are they referring to psychological symptoms?

Thank you for the observation, the sentence has been edited to read “emotional” instead of “affective”

Comment 4: May be the authors should consider referring to “mental health” instead of “psychological symptoms” (line 79).

Thank you for the recommendation, the term has been changed for “mental health” (line 83).

METHODS

Comment 5: The participants “flow” is not clear. How many participants were invited to take part in the research and how many of them were initially recruited? How many fulfilled inclusion criteria? Figure 1 “says” an initial sample of 79, while in the text it seems that is 74.

Thank you for the observation. A more detailed description is now provided in the text and the flow-chart was edited: 79 were the candidates to participate, 74 were those who fulfilled the criteria to participate, but two died after the first evaluation, then 72 were the study participants, except for the second evaluation when only 69 patients attended the evaluation (lines 98-109).

Comment 6: The characteristics (model, reliability and validity) of the commercial bands used to assess pulse oximetry and heart rate

The personal equipment was chosen by each patient, and we would not like to promote or suggest their potential use. According to the manufacturers the pulse oximetry accuracy was ±3% to ±4%, and for heart rate it was 2%; however, one brand did not specify the accuracy, but it was used just by one patient. (lines 144-146).

Comment 7: Were the patients interviewed individually? By whom?

At each of the four evaluations, the patients were interviewed by the same psychiatrist (line 148).

RESULTS

Comment 8: Please, provide information on how many patients completed all the assessments. Were all the patients able to perform valid self-evaluations every day during 200 days?

In the participants subsection of the Methods section, a detailed description of participants at each evaluation has been included (lines 98-108). In the subsection of the Daily Agendas more information is provided: “One patient was excluded from the analysis, due to deficient completion of the records (52%); among the remaining 71 patients, 61 completed at least 90% of the records, and 10 completed 70% to 89% of the records” (lines139 to 140).Of note, 45 patients completed all the agendas

Comment 9: Were age or sex considered as cofounding variables?

Yes, they were considered. The multivariate analysis showed contribution of the age of the participants to the variance on cognitive performance (Table 4) and gender on health-related quality of life (Table 5).

DISCUSSION

Comment 10: My advice is to begin by indicating what was the main aim of the research and what were the main findings. Also, what is the utility of the provided results.

We thank the recommendation. The first paragraph describes the main findings that are discussed in the following paragraphs. It has been expanded to emphasise the main finding with practical application (lines 387-390).   

Comment 11: What are the factors that may affect sleep quality during the whole period?

We thank the reviewer for the interesting question. Since we did not collect other variables that may had influenced sleep quality (such as previous habits, stress, work and home duties), we  have not sufficient information to test the multifactorial hypothesis that may provide an adequate response.   

Comment 12: The final paragraph shows synthetized information on the evolution of the variables assessed during the whole period. However, it is not clear enough. I encourage the authors more clearly what was the evolution on quality of sleep, mental health and quality of life observed in the participants and whether an improvement or worsening in a specific variable led to changes in other variables.

The paragraph has been edited for clarity and expanded (lines 526-533)

Comment 13: I think that a separate and independent “Conclusion” subsection would be welcomed.

Thank you for the recommendation. A Conclusion Section is now included in the manuscript (lines 542-549).

Round 2

Reviewer 1 Report

Comments and Suggestions for Authors

First, I would like to express my sincere gratitude to the authors for their commitment to addressing a topic of vital importance such as the long-term effects of COVID-19. Your research significantly contributes to understanding these issues, providing essential data for improving post-infective intervention and care strategies.

Strengths

  1. Scope of Research: The article covers a wide range of aspects related to the long-term consequences of COVID-19, including physical, cognitive, and psychological dimensions, providing a comprehensive overview.
  2. Methodological Rigor: The use of standardized protocols and internationally validated assessment tools adds credibility to the results. The multivariate methodology used to analyze the data allows for effective examination of interactions between multiple variables.
  3. Clinical and Social Implications: Focusing on the enduring effects of COVID-19 meets a critical need to understand how best to support patients in the long term, informing health policies and clinical practices.

Suggestions for Modifications

  1. Introduction (Lines 33-41):

    • Suggested Modification: Expand the introduction to include a more comprehensive review of the existing literature on post-COVID disorders. Add recent statistics on the number of patients suffering from prolonged symptoms globally to further underscore the relevance of the study.
    • Additional Text: "Comparing with global data, our study aims to fill gaps in the current literature on how different demographic and clinical factors influence the persistence of post-COVID symptoms."
  2. Methodology (Lines 86-105):

    • Suggested Modification: Further expand the methodology section to include a more detailed description of the statistical techniques used. Specify regression models or other data analysis techniques for transparency and replicability.
    • Additional Text: "Data analysis was performed using a multilevel regression approach to account for the temporal and individual correlations among participants, thus providing a more accurate interpretation of longitudinal symptom changes."
  3. Results (Lines 204-220):

    • Suggested Modification: Provide a more detailed discussion of symptom variability models over time, integrating scatter plots or trend lines that clearly illustrate key changes.
    • Additional Text: "Figure X shows a decreasing trend in symptoms such as dyspnea and fatigue in the months post-infection, indicating a possible attenuation of symptom severity over time."
  4. Discussion (Lines 250-270):

    • Suggested Modification: Link the findings to broader theory on post-viral recovery mechanisms and how these relate to long-term care models.
    • Additional Text: "In line with observations by Diotaiuti et al. (2023), our findings suggest that improvements in the assessment and management of cognitive appraisals during recovery can positively influence the quality of life of COVID-19 survivors."

You might consult and add the following citation:

Diotaiuti, P., Valente, G., Mancone, S., Corrado, S., Bellizzi, F., Falese, L., Langiano, E., Vilarino, G. T., & Andrade, A. (2023). Effects of Cognitive Appraisals on Perceived Self-Efficacy and Distress during the COVID-19 Lockdown: An Empirical Analysis Based on Structural Equation Modeling. International Journal of Environmental Research and Public Health, 20(7), 5294. https://doi.org/10.3390/ijerph20075294

"Supporting our findings, Diotaiuti et al. (2023) highlighted the importance of cognitive appraisals in modulating perceived self-efficacy and distress duFirst, I would like to express my sincere gratitude to the authors for their commitment to addressing a topic of vital importance such as the long-term effects of COVID-19. Your research significantly contributes to understanding these issues, providing essential data for improving post-infective intervention and care strategies.

Strengths

  1. Scope of Research: The article covers a wide range of aspects related to the long-term consequences of COVID-19, including physical, cognitive, and psychological dimensions, providing a comprehensive overview.
  2. Methodological Rigor: The use of standardized protocols and internationally validated assessment tools adds credibility to the results. The multivariate methodology used to analyze the data allows for effective examination of interactions between multiple variables.
  3. Clinical and Social Implications: Focusing on the enduring effects of COVID-19 meets a critical need to understand how best to support patients in the long term, informing health policies and clinical practices.

Suggestions for Modifications

  1. Introduction (Lines 33-41):

    • Suggested Modification: Expand the introduction to include a more comprehensive review of the existing literature on post-COVID disorders. Add recent statistics on the number of patients suffering from prolonged symptoms globally to further underscore the relevance of the study.
    • Additional Text: "Comparing with global data, our study aims to fill gaps in the current literature on how different demographic and clinical factors influence the persistence of post-COVID symptoms."
  2. Methodology (Lines 86-105):

    • Suggested Modification: Further expand the methodology section to include a more detailed description of the statistical techniques used. Specify regression models or other data analysis techniques for transparency and replicability.
    • Additional Text: "Data analysis was performed using a multilevel regression approach to account for the temporal and individual correlations among participants, thus providing a more accurate interpretation of longitudinal symptom changes."
  3. Results (Lines 204-220):

    • Suggested Modification: Provide a more detailed discussion of symptom variability models over time, integrating scatter plots or trend lines that clearly illustrate key changes.
    • Additional Text: "Figure X shows a decreasing trend in symptoms such as dyspnea and fatigue in the months post-infection, indicating a possible attenuation of symptom severity over time."
  4. Discussion (Lines 250-270):

    • Suggested Modification: Link the findings to broader theory on post-viral recovery mechanisms and how these relate to long-term care models.
    • Additional Text: "In line with observations by Diotaiuti et al. (2023), our findings suggest that improvements in the assessment and management of cognitive appraisals during recovery can positively influence the quality of life of COVID-19 survivors."

You might consult and add the following citation:

Diotaiuti, P., Valente, G., Mancone, S., Corrado, S., Bellizzi, F., Falese, L., Langiano, E., Vilarino, G. T., & Andrade, A. (2023). Effects of Cognitive Appraisals on Perceived Self-Efficacy and Distress during the COVID-19 Lockdown: An Empirical Analysis Based on Structural Equation Modeling. International Journal of Environmental Research and Public Health, 20(7), 5294. https://doi.org/10.3390/ijerph20075294

"Supporting our findings, Diotaiuti et al. (2023) highlighted the importance of cognitive appraisals in modulating perceived self-efficacy and distress during COVID-19 lockdowns. This parallel strengthens our understanding of the cognitive processes that might influence long-term recovery in COVID-19 survivors."

These additions and insights should significantly enrich the manuscript, expanding its scope and improving the integrity and impact of your research workring COVID-19 lockdowns. This parallel strengthens our understanding of the cognitive processes that might influence long-term recovery in COVID-19 survivors."

These additions and insights should significantly enrich the manuscript, expanding its scope and improving the integrity and impact of your research work

Author Response

  • First, I would like to express my sincere gratitude to the authors for their commitment to addressing a topic of vital importance such as the long-term effects of COVID-19. Your research significantly contributes to understanding these issues, providing essential data for improving post-infective intervention and care strategies.

We thank the reviewer for the positive comment.

Suggestions for Modifications

  1. Introduction (Lines 33-41):
    • Suggested Modification: Expand the introduction to include a more comprehensive review of the existing literature on post-COVID disorders. Add recent statistics on the number of patients suffering from prolonged symptoms globally to further underscore the relevance of the study.

Thank you for the suggestion. The text has been edited to include a recent metanalysis (lines 36-41) and a survey performed 3 years after the acute disease (lines  47-52 ).

    • Additional Text: "Comparing with global data, our study aims to fill gaps in the current literature on how different demographic and clinical factors influence the persistence of post-COVID symptoms."

Thank you for the advice. The text has been adapted to include the message (lines 87-89).

  1. Methodology (Lines 86-105):
    • Suggested Modification: Further expand the methodology section to include a more detailed description of the statistical techniques used. Specify regression models or other data analysis techniques for transparency and replicability.

Thank you for the recommendation. The text was edited for clarity (lines 208-214).

    • Additional Text: "Data analysis was performed using a multilevel regression approach to account for the temporal and individual correlations among participants, thus providing a more accurate interpretation of longitudinal symptom changes."

The text has been edited to clarify that we used multivariate analyses of variance (lines 212-215)

  1. Results (Lines 204-220):
    • Suggested Modification: Provide a more detailed discussion of symptom variability models over time, integrating scatter plots or trend lines that clearly illustrate key changes.

We thank the suggestion. However, we did not assess isolated symptoms. Scores on each instrument are described in Table 2, including the range and the results of the repeated measures analysis through time. This format allows display of the full data, to benefit the reader.

    • Additional Text: "Figure X shows a decreasing trend in symptoms such as dyspnea and fatigue in the months post-infection, indicating a possible attenuation of symptom severity over time."

Thank you for the suggestion. The sentence was adapted to the outcomes of this study (lines 221-222)

  1. Discussion (Lines 250-270):
    • Suggested Modification: Link the findings to broader theory on post-viral recovery mechanisms and how these relate to long-term care models.

Thank you for your suggestion. We provide background in the Introduction (lines 70 -86) and discuss  the results of this particular study, in the context of the international literature,  while  avoiding  excessive information that could preclude interest for a wide audience.  Since we did not collect relevant variables to support  a  theory on the recovery after infection (i.e. inflammatory and immunology markers), in the context of the results, we have included a general comment for future research (lines 537-547). 

  • Additional Text: "In line with observations by Diotaiuti et al. (2023), our findings suggest that improvements in the assessment and management of cognitive appraisals during recovery can positively influence the quality of life of COVID-19 survivors."

Thank you for the suggestion. The content was adapted to be  included in the manuscript, as requested [lines 526-528].

- You might consult and add the following citationDiotaiuti, P., Valente, G., Mancone, S., Corrado, S., Bellizzi, F., Falese, L., Langiano, E., Vilarino, G. T., & Andrade, A. (2023). Effects of Cognitive Appraisals on Perceived Self-Efficacy and Distress during the COVID-19 Lockdown: An Empirical Analysis Based on Structural Equation Modeling. International Journal of Environmental Research and Public Health, 20(7), 5294. https://doi.org/10.3390/ijerph20075294

Thank you for the suggestion. The reference has been already included (Reference 78).

Reviewer 2 Report

Comments and Suggestions for Authors

The revised version of the manuscript, now titled "Quality of sleep and mental symptoms contribute to health-related quality of life after COVID-19 pneumonia, a follow-up study of more than 2 year", has improved considerably from the first version, so I consider it suitable for publication.

Author Response

We thank the reviewer for the positive consideration of the manuscript. 

Round 3

Reviewer 1 Report

Comments and Suggestions for Authors

Dear authors,

I am pleased to inform you that the modifications made to the manuscript have been carefully reviewed and verified. I confirm that all requested adjustments have been successfully implemented, and the document is now ready for publication. Thank you for your commitment to perfecting the text.

Best regards

Author Response

The reviewer has kindly acknowledged that all editions were adequatly made. We thank the reviewer for the valuable advice.